# Paternal-age-related de novo mutations and risk for five disorders

Jacob L. Taylor[1,2,3], Jean-Christophe P.G. Debost[4,5,6], Sarah U. Morton[7], Emilie M. Wigdor [2,3,8], Henrike O. Heyne[2,3,8], Dennis Lal[2,8,9,10], Daniel P. Howrigan[2,8], Alex Bloemendal[2,3,8], Janne T. Larsen [4,5], Jack A. Kosmicki [2,3,8,11], Daniel J. Weiner[2,3,8], Jason Homsy[12], Jonathan G. Seidman[12], Christine E. Seidman [12,13,14], Esben Agerbo [4,5,15], John J. McGrath [4,16,17], Preben Bo Mortensen[4,5,15,18], Liselotte Petersen[4,5], Mark J. Daly [2,3,8] & Elise B. Robinson[2,3,8,19]

There are established associations between advanced paternal age and offspring risk for psychiatric and developmental disorders. These are commonly attributed to genetic mutations, especially de novo single nucleotide variants (dnSNVs), that accumulate with increasing paternal age. However, the actual magnitude of risk from such mutations in the male germline is unknown. Quantifying this risk would clarify the clinical significance of delayed paternity. Using parent-child trio whole-exome-sequencing data, we estimate the relationship between paternal-age-related dnSNVs and risk for five disorders: autism spectrum disorder (ASD), congenital heart disease, neurodevelopmental disorders with epilepsy, intellectual disability and schizophrenia (SCZ). Using Danish registry data, we investigate whether epidemiologic associations between each disorder and older fatherhood are consistent with the estimated role of dnSNVs. We find that paternal-age-related dnSNVs confer a small amount of risk for these disorders. For ASD and SCZ, epidemiologic associations with delayed paternity reflect factors that may not increase with age.

[1] Department of Psychiatry, Brigham and Woman's Hospital, Boston, MA 02115, USA. [2] Stanley Center for Psychiatric Research, Broad Institute of MIT and Harvard, Cambridge, MA 02142, USA. [3] Program in Medical and Population Genetics, Broad Institute of MIT and Harvard, Cambridge, MA 02412, USA. [4] National Centre for Register-based Research, Department of Economics and Business, Aarhus University, Aarhus 8210, Denmark. [5] The Lundbeck Foundation Initiative for Integrative Psychiatric Research, iPSYCH, Aarhus 8210, Denmark. [6] Aarhus University Hospital, Risskov, Department P, Aarhus 8200, Denmark. [7] Division of Newborn Medicine, Boston Children's Hospital and Harvard Medical School, Boston, MA 02115, USA. [8] Analytic and Translational Genetics Unit, Department of Medicine, Massachusetts General Hospital and Harvard Medical School, Boston, MA 02114, USA. [9] Cologne Center for Genomics, University of Cologne, Cologne 50923, Germany. [10] Psychiatric and Neurodevelopmental Genetics Unit, Department of Psychiatry, Massachusetts General Hospital, Boston, MA 02114, USA. [11] Program in Genetics and Genomics, Biological and Biomedical Sciences, Harvard Medical School, Boston, MA 02115, USA. [12] Department of Genetics, Harvard Medical School, Boston, MA 02115, USA. [13] Cardiovascular Division, Brigham and Women's Hospital, Boston, MA 02115, USA. [14] Howard Hughes Medical Institute, Harvard Medical School, Boston, MA 02115, USA. [15] Centre for Integrated Register-based Research, CIRRAU, Aarhus University, Aarhus 8210, Denmark. [16] Queensland Brain Institute, The University of Queensland, Brisbane 4072 Queensland, Australia. [17] Queensland Centre for Mental Health Research, The Park Centre for Mental Health, Richlands 4076 Queensland, Australia. [18] Department of Biomedicine and iSEQ, Centre for Integrative Sequencing, Aarhus University, Aarhus 08210, Denmark. [19] Department of Epidemiology, Harvard T.H. Chan School of Public Health, Boston, MA 02115, USA. Correspondence and requests for materials should be addressed to E.B.R. (email: erobinso@hsph.harvard.edu)

Epidemiologic associations between advanced paternal age and increased offspring risk is particularly well characterized in autism spectrum disorder (ASD) and schizophrenia (SCZ)[1–7] but has also been found in other disorders including congenital heart disease (CHD)[8], epilepsy[9] and intellectual disability (ID)[10]. These associations are commonly attributed to age-related de novo mutations, especially de novo single nucleotide variants (dnSNVs). These mutations arise in paternal germ cells 3–4 times more often than they do in maternal germ cells[11,12]. As men age, dnSNVs are observed to increase in their offspring[13]. Paternal-age-related de novo variants are therefore likely to increase risk for any dnSNV-influenced disorder. However, the actual magnitude of risk conferred by such mutations is unknown[14] and could in fact be far smaller than that suggested by the epidemiologic associations.

The possibility that the ASD and SCZ associations may not be driven primarily by accumulating de novo variants received support in a recent study using simulated data that found that common polygenic risk, rather than de novo variation, could be most relevant to the observed epidemiologic patterns[15]. These simulated results are consistent with the hypothesis that common, inherited genetic risk for psychiatric disorders may also predict age of childbearing[16,17]. That is, individuals who carry elevated inherited risk for ASD, for example, might on average have children later in life[1]. It is now possible to quantify the influence of paternal-age-related de novo variants on disease risk using empirical data, as, over the last several years, large parent–child trio whole-exome-sequencing studies have been able to establish the relationship between de novo variation and risk for each of ASD, CHD, ID, neurodevelopmental disorders with epilepsy (EPI) and SCZ[18–33].

The purpose of the present study is to directly estimate risk for each of these five disorders created by paternal-age-related de novo mutations in the exome, the protein-coding portion of the genome. Clarity regarding the proportion of de novo mutation risk conferred by advancing paternal age could influence individual decision-making and genetic counseling[34]. We describe a model to quantify the risk associated with paternal-age related dnSNVs and apply this model to the five disorders (listed above) for which large parent-child trio whole-exome sequencing data currently exist. Using Danish national registry data, we then investigate the degree to which the epidemiologic association between each disorder and advanced paternal age is consistent with the estimated role of de novo mutations. In the cases of CHD, ID and EPI, epidemiologic estimates of paternal age risk are consistent with the magnitude of the effect attributable to dnSNVs. However, epidemiologic effects for ASD and SCZ significantly exceed the risk that could be explained by dnSNVs alone. While increasing dnSNVs due to advanced paternal age confer a small amount of offspring risk for psychiatric and developmental disorders, the epidemiologic associations of ASD and SCZ with delayed paternity largely reflect factors that cannot be assumed to increase with age.

## Results

**Paternal-age-related dnSNV risk model.** We describe a statistical model that, for fathers of any two given ages, compares offspring disease risk due to nonsynonymous (missense and protein truncating) dnSNVs. For the remainder of the text, dnSNV will be used to reference nonsynonymous dnSNVs, as they are the de novo variant types consistently associated with risk for human disease[18–33,35]. The model incorporates two relationships: (1) the general population relationship between paternal age and dnSNV accumulation, and (2) the case-control relationship between dnSNVs and an outcome of interest

(e.g. SCZ). To estimate the first, we used data from the healthy siblings of ASD probands in the Simons Simplex Collection (SSC)[26,36]. To estimate the second, we used published studies of ASD, CHD, EPI, ID, and SCZ[18–33].

The statistical model for predicting changes in population-level disease incidence due to paternal age related dnSNVs in the exome is:

$$\frac{INC_{age2}}{INC_{age1}} = \frac{INC_{baseline}(OR_{nonsyn}^{\#nonsyn(age2)})}{INC_{baseline}(OR_{nonsyn}^{\#nonsyn(age1)})} = \frac{\left(OR_{PTV}^{\#PTV(age2)}\right)*\left(OR_{missense}^{\#missense(age2)}\right)}{\left(OR_{PTV}^{\#PTV\,age1}\right)*\left(OR_{missense}^{\#missense(age1)}\right)}$$

$$(1)$$

where $\#nonsyn(age)$ = number of dnSNVs expected in offspring of fathers of a particular age = $e^{i+age^{\star}\beta}$ (where $i$ is the intercept, and $\beta$ the coefficient of a Poisson regression model in which number of dnSNVs in offspring is regressed against paternal age at birth; see Methods). $OR_{nonsyn}/OR_{PTV}/OR_{missense}$ = odds ratio reflecting disease risk associated with having one additional nonsynonymous, protein truncating or missense variant respectively.

Its development is described in detail in the Methods. The output of the model is an incidence rate ratio (IRR) reflecting the increase in dnSNV-related disease risk in offspring of older fathers compared with offspring of younger fathers. In brief, one can estimate disease incidence in offspring of fathers of a specified age (e.g. people born when their fathers were 25) by multiplying (a) disease incidence among individuals who carry zero dnSNVs ($INC_{baseline}$) and (b) the odds ratio (OR) associated with one additional nonsynonymous dnSNV, exponentiated to the expected number of dnSNVs in offspring of fathers of the specified age (see Methods, below). Baseline incidence can be unknown or variable with respect to time as, algebraically, it is not needed for estimating the IRR. As noted in Equation (1), effect size variation across different types of dnSNVs is also algebraically irrelevant under the assumption that all types of dnSNVs increase with age at the same rate (Supplementary Notes 1 and 2).

To show this we use the example of missense and protein truncating variants (PTVs), which have different average effect sizes on risk for ASD. This logic also applies to post-zygotic mutations (PZM) that are confused for de novo variants. PZMs would not affect our estimates under the assumption that PZM rate is unassociated with paternal age (Supplementary Note 3).

As described in Supplementary Notes 4 and 5 and Supplementary Fig. 1, the model's estimates are also robust to: plausible variation in the expected (control) rate of dnSNVs, plausible variation in the estimated effect size of case dnSNVs, and the inclusion of de novo copy number variants. The model assumes that risk-conferring statistical interaction among dnSNVs do not commonly occur and additionally assumes, contrary to the selfish spermatogonial selection model[37], that de novo variants that emerge in sperm cells later in life are no more (or less) likely to be disease-associated than other de novo variants. In Supplementary Note 6 we show that interactions among disease-associated dnSNVs are not currently identifiable in the ASD data and discuss why they are unlikely to be relevant for any of the other disorders. In supplementary Note 7 we discuss the selfish spermatogonial selection theory, discuss the implications for our model if such a mechanism were significant, and explain why this phenomenon is unlikely to substantially influence our findings.

Because not every cohort used the same technology for sequencing individuals or the same procedure for calling dnSNVs, we adjusted dnSNV rate in each cohort by its rate of synonymous de novo variation (See Methods, below). We did so based on the

**Table 1 The burden of dnSNVs in five disorders**

| Disorder | N trios | nonsynonymous dnSNVs per person | Synonymous variants per person | Unadjusted OR (p) | Adjusted OR (p) | Refs |
|---|---|---|---|---|---|---|
| Intellectual disability | 5264 | 1.16 | 0.29 | $1.78\ (1 \times 10^{-75})$ | $1.50\ (3 \times 10^{-11})$ | 2–7 |
| Neurodevelopmental disorders with epilepsy | 1942 | 1.09 | 0.18 | $1.59\ (4 \times 10^{-40})$ | $1.48\ (5 \times 10^{-6})$ | 8 |
| Congenital heart disease | 2645 | 0.92 | 0.27 | $1.41\ (5 \times 10^{-23})$ | $1.31\ (9 \times 10^{-5})$ | 9 |
| Autism spectrum disorders | 2508 | 0.79 | 0.25 | $1.22\ (4 \times 10^{-8})$ | 1.20 (0.009) | 10 |
| Schizophrenia | 1077 | 0.72 | 0.22 | 1.11 (0.02) | 1.22 (0.03) | 11–17 |
| Control | 1902 | 0.65 | 0.25 | — | — | 10 |
| Control group for EPI | 1911 | 0.69 | 0.17 | — | — | 8 |

The burden of dnSNVs in five phenotypes. Nonsynonymous dnSNVs per person and synonymous variants per person refer to the number of de novo variants per proband across all trio families. Unadjusted OR is the rate of nonsynonymous dnSNVs per person in affected probands divided by the same rate in Simons Simplex Collection control siblings. The adjusted OR reflects the same ratio, where the rate of nonsynonymous dnSNVs per person in affected probands is adjusted by a factor which would equalize the synonymous variant rate across cases and controls. The method for generating p-values is described in the methods. Because nonsynonymous dnSNVs were called using a different method for the EPI probands compared with all other probands, there is a distinct control dnSNV rate for comparison with EPI

assumption that the true rate of synonymous variation across cohorts is approximately equal. In Supplementary Note 8 we show that the results we describe here are robust to whether or not one accepts this assumption (Supplementary Figs 2 and 3; Supplementary Tables 1 and 2). Finally, in Supplementary Note 9 we show that the results are not substantially affected by modeling dnSNV accumulation to begin at puberty (approximated by age 13).

**dnSNVs accumulate with advancing paternal age**. Using the 1827 SSC control trios in whom data on paternal age was available, we found that dnSNVs accumulate with advancing paternal age at a rate of 3.1% per year ($p = 2 \times 10^{-10}$, Poisson regression; see Supplementary Note 10 for a discussion of the fact that the model treats dnSNVs as increasing proportionally, rather than linearly, with increasing paternal age). This estimate is consistent with those produced by recent whole genome sequencing studies (Supplementary Table 3), and did not vary by type of variant examined (Supplementary Fig. 4). Table 1 summarizes each data set used to estimate the risk associated with one additional dnSNV for each disorder. For each disorder, cases had an excess of dnSNVs compared with controls ($p < 0.05$, exact Poisson tests), both in absolute terms, and when adjusting for rate of synonymous variation.

**Risk from paternal-age-related dnSNVs**. Figure 1 illustrates the impact of paternal-age-related dnSNVs on offspring risk for each of the five disorders. For offspring of a 45-year-old father compared with offspring of a 25-year-old father, the IRR of each disorder ranged from: 1.09 for ASD and SCZ to 1.20 for ID (Supplementary Table 4). This translates to a ~10–20% increase in risk over that paternal age span. This increase in risk must be interpreted against the low overall incidences of each of the five disorders[22,24,38–40]. For example, in a condition with 1% baseline incidence, a 20% increase in risk would translate to a 1.2% probability of the outcome.

**Comparing dnSNV-based risk with epidemiological observation**. We conducted a companion analysis in the population-based Danish national patient registry to (1) estimate the epidemiologic association between advanced paternal age and risk for each disorder, and (2) compare the epidemiologic associations to those from our dnSNV model. Previous reports have used the Danish registries to estimate epidemiologic paternal age effects for ASD, schizophrenia and CHD[1,41]. Our approach is similar to these earlier efforts, but modified slightly to maximize

concordance with our dnSNV model (See Methods, below). In brief, we compared risk for an ICD-10 diagnosis of each of the five disorders between Danish children born to fathers 20–29 versus > 39. All births occurred between 1955 and 2012. As described in detail in the Methods, the definition of cases and controls differed for the CHD analyses, to be consistent with the trio sequencing studies' requirement that children be diagnosed by one year of age. Odds ratios (ORs) were accordingly used to estimate CHD risk associated with paternal age, while hazard ratios (HRs) were used for all other disorders. Both ORs and HRs are functionally equivalent to IRRs[42] for disorders with low cumulative incidence (e.g. <5%) and risk factors with small effect sizes[43].

Figure 2 presents our estimates of paternal age risk derived through the dnSNV and epidemiologic models, specifically comparing risk for each disorder between fathers older than 39 versus those in their 20s. In the epidemiologic model, the HR (or OR) for each disorder ranged from: 1.06 (0.89–1.18) for CHD to 1.68 (1.51–1.86) for ASD (Supplementary Table 5).

We input the mean paternal ages of the Danish fathers into the dnSNV model, and ran the dnSNV model for each disorder. In the Danish cohort used to estimate the epidemiologic association between advanced paternal age and ASD, SCZ, ID and EPI, the mean ages of fathers in their 20s and over 39 were 26.2 and 44.1, respectively. For the cohort used to estimate the epidemiologic association between advanced paternal age and CHD, the mean ages were 27.1 and 43.7, respectively. For CHD and EPI the estimates from the epidemiologic and dnSNV models are statistically indistinguishable ($p = 0.50$ and $p = 0.27$ respectively, empiric p value as described in Methods; Supplementary Table 5). This means that we do not observe a paternal age effect in the population outside the bounds of what could be explained by paternal-age-related de novo variation. For ASD and SCZ, however, the HRs derived from the epidemiologic data were significantly greater than the IRRs expected from paternal-age-related dnSNVs ($p = 2 \times 10^{-5}$ for ASD and $p = 0.02$ for SCZ, empiric p value as described in Methods).

The association between advanced paternal age and ID risk in the Danish population may exceed that which can be accounted for by dnSNVs ($p = 0.05$, empiric $p$ value, as described in Methods). Minor variation in the statistical comparison is induced by control for synonymous rate, as described below.

These results suggests that much of the paternal age effect observed in the population for ASD, and most likely for SCZ, is attributable to factors other than de novo mutations, consistent with the findings of Gratten et al.[15]. The risk for ASD associated with advance paternal age in the population was nearly an order

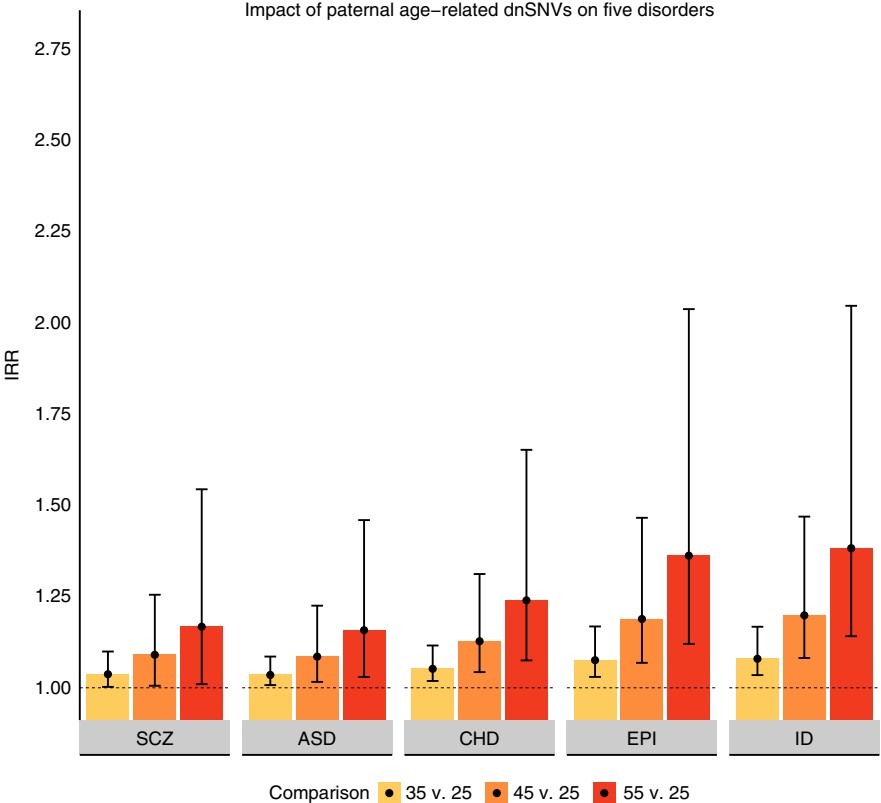

**Fig. 1** Impact of paternal age-related dnSNVs on five disorders. IRR Incidence rate ratio, SCZ schizophrenia, ASD autism spectrum disorder, CHD congenital heart disease, EPI epilepsy, ID intellectual disability. Error bars reflect 95% confidence intervals. Note that confidence intervals are not valid for comparing dnSNV effect within or across disorders (See Methods)

of magnitude (×9.3) greater than that which our model could attribute to dnSNVs. Further, the epidemiologic effect observed for ASD was greater than the epidemiologic effect observed for intellectual disability ($p = 0.011$, z test comparing log hazard ratios using associated standard errors). As ID collections have a substantially greater observed rate of de novo variants than ASD collections (see Table 1), these data very strongly suggest that factors other than de novo variation drive the association between paternal age and autism risk observed in the population.

### Discussion
In this analysis, we directly estimated the extent to which paternal-age-related de novo mutations create offspring disease risk for ASD, CHD, EPI, ID, and SCZ. For each disorder, we show that the causal effect attributable to de novo mutations in the exome is small. By delaying paternity from his mid-20s to his mid-40s, for example, a man's offspring would be only about 9% more likely to develop schizophrenia, and 20% more likely to develop ID, consequent to his age-related de novo mutations. Against the low prevalence of each condition, this increase in risk results in a small predicted increase in incidence at the population level. Genetic counselors and others can use these data to inform discussions with patients about risks associated with delayed parenting.

Our estimates could be affected by several additional properties of the data and analysis. First, the trio-sequencing results for four of the disorders (ASD, ID, EPI and CHD) came from data sets that are partially overlapping. These disorders are highly comorbid as diagnosed, but it is possible that the dnSNV estimates we provide for these disorders are more similar to each

other than they would be if totally independent cohorts were used. This limitation cannot be addressed in the present analysis as we employed published summary data from each cohort. Similarly, none of the cohorts used are random samples of individuals diagnosed with specific disorders. It is possible that the different ascertainment strategies used for each cohort lead to subtly different rates of dnSNVs. This is particularly possible in the case of disorders with substantial phenotypic and etiologic heterogeneity, like ASD. Third, while we have explicitly considered the role of de novo mutations in the exome, it is possible that de novo single nucleotide variants in the rest of the genome could create additional risk associated with advanced paternal age. However, there is currently no evidence of which we are aware for enrichment of de novo variants outside of the exome in genetically complex human diseases[35]. More importantly, if de novo variants that do not effect protein structure did contribute sufficiently to some genetically complex disorders (ASD, SCZ) to cause a large increase in their epidemiologic association with advanced paternal age, it would be difficult to account biologically for the fact that the mutational burden in the exome explains the association between advanced paternal age other genetically complex disorders (CHD, EPI and ID) as we show here.

Another possible limitation is that we did not include maternal age as a covariate in the model, as paternal and maternal ages were too highly correlated ($r = 0.72$) to distinguish between their effects (Supplementary Note 11), and the SSC de novo variant data have not been phased in adequate fraction for the effects to be estimated separately[26]. There is evidence that dnSNVs increase in egg cells as maternal age increases and that certain genomic regions may be more vulnerable to maternal age- compared with

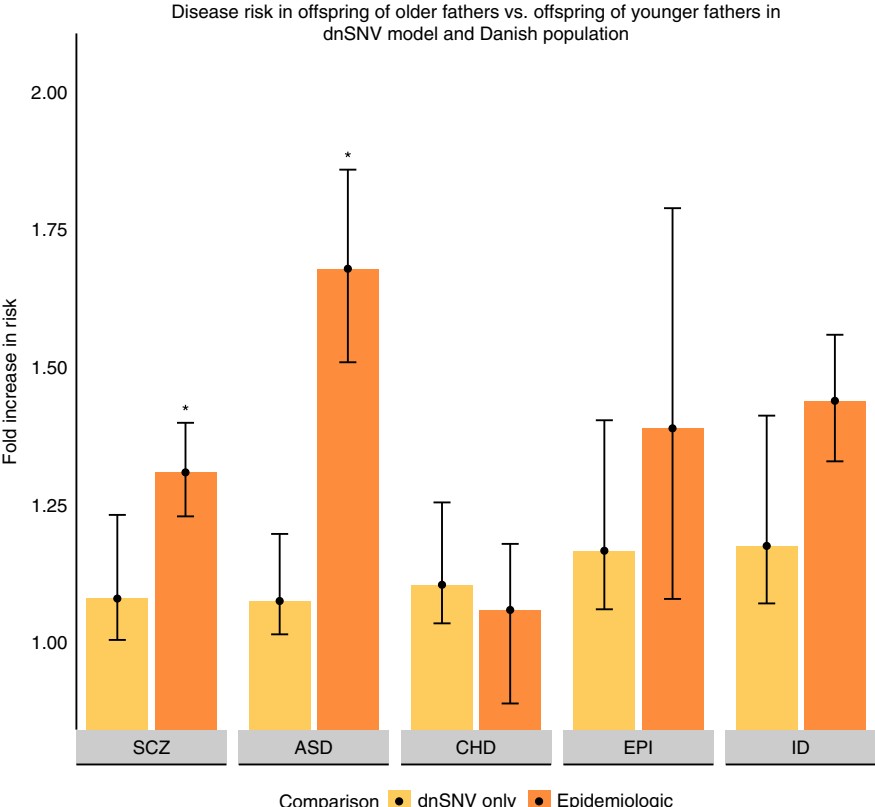

Disease risk in offspring of older fathers vs. offspring of younger fathers in dnSNV model and Danish population

Comparison ● dnSNV only ● Epidemiologic

**Fig. 2** Disease risk in offspring of older fathers vs. offspring of younger fathers in dnSNV model and Danish population. SCZ schizophrenia, ASD autism spectrum disorder, CHD congenital heart disease, EPI neurodevelopmental disorders with epilepsy, ID intellectual disability. Error bars reflect 95% confidence intervals. *$p < 2 \times 10^{-4}$ against null hypothesis that epidemiologic association between advanced paternal age and disease risk is equivalent to the dnSNV model's estimate (empiric $p$ value as described in Methods)

paternal age-associated mutations[44]. There is also evidence that the rate at which dnSNVs accumulate with maternal age, may itself increase with age[45]. However, it is also clear that the maternal age effect on dnSNV accumulation is much smaller than the paternal age effect[11,12]. For this reason, and because paternal and maternal ages are so highly correlated, failing to separately model the impact of maternal age are not likely to substantially change our results. The modeling approach used here results in estimates of paternal-age risk that include (and are increased by) the correlated risks associated with maternal age. As the model is designed to predict changes in disease incidence at a population-level, the results presented here will best generalize to populations in which maternal and paternal age have a similar relationship to that observed in the United States.

The average age of childbearing has increased across many communities and cultures. These results suggest that these trends are unlikely to lead to a substantial increase in common human diseases through the accumulation of dnSNVs (Supplementary Note 12).

## Methods

**Ethical approval.** All analyses described were approved by the Partners Healthcare Institutional Review Board protocol number 2015P002376. Individuals contributing data through the SSC were consented as described previously[36]. No individual-level data from other whole-exome sequenced trio cohorts was used. Informed consent is therefore not applicable.

The registry-based analyses were approved by the Danish Data Protection Agency. Studies that utilize this resource, but that do not involve recontacting individuals do not require individualized informed consent[46].

**Developing the dnSNV model.** As noted above the model incorporates two relationships: (1) the relationship between paternal age and dnSNV accumulation, and (2) the relationship between dnSNVs and an outcome of interest (e.g. ASD).

We first empirically estimated the general population relationship between paternal age and number of dnSNVs in offspring. The analysis used data from the largest existing sample of control individuals with available whole-exome dnSNV data: the healthy siblings of ASD probands in the Simons Simplex Collection (SSC)[26,36]. There were $n = 1821$ such families where data were available on a) parental age at birth of the unaffected sibling and b) dnSNV count in the unaffected sibling. Using Poisson regression (implemented through the glm function in R), we regressed the number of dnSNVs in each unaffected sibling of ASD probands on paternal age at birth. The intercept ($i = -1.45$) from this model represents the natural logarithm of the expected number of dnSNVs in offspring of (theoretical) fathers aged zero. The coefficient ($\beta = 0.031$) represents the proportionate increase in number of dnSNVs for each year of increased paternal age (Equation (1); Supplementary Fig. 5).

We next estimated the relationship between dnSNVs and offspring risk for each of ASD, CHD, EPI, ID, and SCZ – the disorders for which large-scale whole-exome-sequenced trio data currently exist. All such data has been published[18–33]. From each study, or group of studies, we extracted the average number of dnSNVs per case. In order to adjust for differing rates of synonymous variation we multiplied the number of nonsynonymous variants in each disease cohort by a factor that would cause the rate of synonymous variants in each cohort to equal the rate in the SSC siblings (Supplementary Table 6).

Among the $n = 1902$ SSC siblings for whom we had data on number of dnSNVs, there are 0.25 de novo synonymous variants per person. We used this rate to adjust the dnSNV rate for ASD, SCZ, CHD, and ID. Due to a technical difference in how variants were called in the EPI probands compared with the other disease outcomes (Supplementary Note 13) we used the observed de novo synonymous rate published in Heyne et al.[24] of 0.16 to adjust the dnSNV rate for EPI.

As an example, among the 2645 probands with CHD, there were 701 de novo synonymous variants for a rate of 0.27 (701/2645) variants per probands. There were 2431 dnSNVs observed among this population. For our primary analysis, we accordingly used a value of 2625 (2431 × 0.27/0.25) variants. An alternative analysis

in which we do not adjust for differing rates of de novo synonymous variation is described in Supplementary Note 8.

Algebraically, the odds ratio (OR) for each disorder associated with having one additional dnSNV, can be described as: $A * D / B * C$ where:

A = rate of dnSNVs in cases

B = rate of dnSNVs in controls

C = rate of de novo synonymous variants in cases

D = rate of de novo synonymous variants in controls

To estimate an average dnSNV rate in controls we again used the healthy siblings of probands from the SSC for the primary analyses ($n = 1902$ for whom we had data on number of dnSNVs). An EPI-specific dnSNV rate was calculated in the SSC siblings using the same calling procedure as the dnSNV rate in the EPI probands[24]. For the EPI analysis, the control dnSNV rate was calculated in $n = 1911$ SSC siblings (Table 1; for further discussion and for other considerations unique to calculating the dnSNV rate in ASD and ID, see Supplementary Notes 14 and 15).

The output of the model presented in Equation (1) is an incidence rate ratio (IRR) reflecting the increase in risk for a given disorder in offspring of older fathers compared to offspring of younger fathers. To determine the incidence of disease in offspring of fathers of a specified age (e.g. people born when their fathers were 25 or people born when their fathers were 45), one starts with the baseline incidence in individuals who carry zero nonsynonymous dnSNVs ($INC_{baseline}$). By definition, the population of individuals with one dnSNV would have incidence equal to $INC_{baseline} * OR_{dnSNV}$, where $OR_{dnSNV}$ is the OR associated with having one additional dnSNV. As each dnSNV is expected to exert its influence independently (see Supplementary Note 6), the incidence in the population of individuals carrying two dnSNV would be $INC_{baseline} * OR_{dnSNV} * OR_{dnSNV} = INC_{baseline} * OR_{dnSNV}^2$.

Following this logic, the disease incidence in the population of individuals carrying n dnSNVs would be $INC_{baseline} * OR_{dnSNV}^n$. Since we have used the unaffected siblings of ASD probands from the SSC to estimate the relationship between paternal age and expected number of dnSNVs, for any given paternal age we can determine the appropriate multiplier to apply to $INC_{baseline}$. Since the goal of the model is to find an IRR comparing risk in offspring of fathers of two specified ages, the baseline incidence itself cancels out (see Supplementary Notes 1–17 for an example calculation and a more formal derivation of the model).

**Generating confidence intervals for the model**. The inputs to the model include three parameters that are estimated directly using real data. These include i (the natural logarithm of the expected number of dnSNVs in offspring of fathers aged zero in the general population), β (an estimate of the proportionate increase in number of dnSNVs per year of increased paternal age in the general population) and OR (an estimate of the OR associated with the increased risk of having a disease given one additional dnSNV). For the first two of these parameters (i and β), we take the associated standard errors generated by the glm function in R. For both of these parameters we then simulate 10,000 new parameters by selecting random numbers within the normal distribution with mean equal to the estimates of the parameters and standard deviations equal to the standard errors associated with these estimates.

To generate standard errors for the natural log (ln) of the OR associated with a single additional dnSNV for each disorder we used the following equation:

A = rate of dnSNVs in cases

B = rate of dnSNVs in controls

C = rate of de novo synonymous variants in cases

D = rate of de novo synonymous variants in controls

$n_{case}$ = number of cases

$n_{control}$ = number of controls

$$ln(OR) = \sqrt{\frac{1}{A * n_{case}} + \frac{1}{B * n_{control}} + \frac{1}{C * n_{case}} + \frac{1}{D * n_{control}}}$$

We simulated 10,000 estimates of the true value of OR by selecting random numbers within the normal distribution with mean equal to our estimate of $ln(OR)$ and standard deviation equal to $SE(ln(OR))$ and then exponentiating the simulated values.

Using the 10,000 simulated values for i, β and OR, for each examination of risk associated with an older paternal age and a younger paternal age, we run the model described in Equation (1) 10,000 times substituting our simulated values for the parameter values that come directly from the data. We then take the 250th smallest and 250th largest resulting values as our 95% confidence interval for each estimate from the model. These 95% confidence intervals are represented as error bars in Fig. 1 and are contained in Supplementary Table 4.

The confidence intervals illustrate the range of possibilities for each comparison (e.g. offspring ASD risk between fathers aged 45 versus 25). However, some of the same parameters are used to generate estimates within and across disorders. Therefore, these estimates are not independent of one another and the associated confidence intervals are not valid for comparing estimates of the dnSNV to one

another. For example, Fig. 1 makes it appear that the estimate for offspring ASD risk between fathers aged 45 versus 25 is substantially overlapping the risk for fathers aged 55 versus 25. However, our confidence that the risk for offspring of 55 year-olds is higher than the risk for offspring of 45 year-olds is actually quite high, as it is a function of our confidence that more dnSNVs lead to more risk for ASD ($p = 0.009$, two sided z test against the null hypothesis that the log(OR) of the impact of a single dnSNV on ASD risk is equal to zero) and our confidence that dnSNVs increase with older paternal age ($p = 2 \times 10^{-10}$, Poisson regression of number of dnSNVs on paternal age).

**Comparing dnSNV-based risk with epidemiological observation**. We queried the Danish National patient registry for all individuals born in Denmark to Danish-born parents between 1955 and 2012 with a diagnosis of ASD, EPI, ID, or SCZ issued between 1995 and 2012, such that all diagnoses were made using ICD-10 criteria. Individuals with any of the above diagnoses made prior to 1995 were excluded. To be consistent with the exome sequencing analysis, CHD cases were only included if diagnosed within one year of birth, and were therefore born between 1994 and 2011. The controls for the CHD analysis were also all born between 1994 and 2011. For all five disorders, we aimed to identify ICD-10 diagnoses that were as similar as possible to the phenotypes included in the exome sequencing studies listed in Table 1 (see below). For each of the five disorders, all non-cases born within the time period were used as controls ($n > 300,000$ for each analysis).

With age as the primary time scale, we used Cox proportional hazard models to compare offspring risk for ASD, EPI, ID, and SCZ between fathers in two different age bins: 20–29 or >39. Broad age bins were defined to maximize statistical power. We set a lower age bound of 20 because very young parent age is also associated with risk for several of the five disorders, though likely through different mechanisms[1,2].

As in the dnSNV model, we did not control for maternal age in the primary analyses. We repeated the analysis controlling for maternal age and/or calendar period (though not both in the same model due to power considerations), as well as using an alternative approach to ascertaining ASD cases in the registry data. We also used an alternative analytic approach to adjust for differences in prevalence of CHD subtypes between the probands in the trio sequencing studies and the cases ascertained through the Danish registry (Supplementary Notes 18–20; Supplementary Tables 7 and 8). None of these alternative analyses substantively changed the results described above and in Fig. 2. As noted above, we also conducted an analysis in which we use the observed rate of dnSNVs from each cohort without adjusting for rate of de novo synonymous mutations. In this analysis, the dnSNV-based estimate appeared significantly smaller than the epidemiologic estimate of the paternal age effect in SCZ and the difference between the two estimates for ID became unambiguously statistically indistinguishable ($p = 0.23$, empiric p value as described in Methods).

For ASD, EPI, ID and SCZ, the mean age for each bin was calculated among all fathers of children born between 1955 and 2012. For CHD, the mean age for each bin was calculated among fathers of children born between 1994 and 2011 and only cases diagnosed within 1 year of birth were included.

To create the confidence intervals for the dnSNV model in Fig. 2, we generated distributions of 100,000 plausible values of results of the dnSNV model comparing offspring of fathers aged 44.1 to offspring of fathers aged 26.2 (for SCZ, ASD, EPI and ID) and offspring of fathers aged 43.7 to offspring of fathers aged 27.1 (for CHD). We did so using the same procedure described above. We again chose the 2.5th and 97.5th percentiles as our lower and upper bounds for each estimate.

Because the probability distributions for estimates of the natural logarithm of a hazard ratio and the natural logarithm of an odds ratio are normally distributed, 95% confidence intervals for the epidemiologic estimates could be directly estimated using the standard errors of the logarithms of the estimates that came out of the regression models used to generate the epidemiologic estimates from the Danish registry data.

We also used these standard errors to directly generate distributions of 100,000 plausible values for the epidemiologic estimates by first using the "rnorm" function in R to generate 100,000 plausible values for the natural logarithm of the estimates and then exponentiating.

To perform statistical tests evaluating whether it is plausible that the epidemiologic findings could be accounted for only by paternal age-associated dnSNVs, we used the two distributions of 100,000 plausible values for each model to generate empiric p values. Specifically, we counted how many times the nth number in the distribution around the smaller estimate (which was the estimate derived using the dnSNV only model for each disorder except CHDs) exceeded the nth number in the distribution around the larger estimate. We divided this quantity by 100,000 to get a one-sided p-value and doubled it to get 2-sided p values. Following this procedure, we were able to generate exact p-values for each disorder except ASD (Supplementary Table 5). For ASD, plausible values for the dnSNV model never exceeded plausible values for the epidemiologic models across all 100,000 iterations. Therefore, we report this p value as $<2 \times 10^{-5}$.

**Matching phenotypes from trio studies to danish registries**. Tables 2 and 3 list the inclusion and exclusion criteria aimed to phenotypically align the Danish registries and trio study data.

**Table 2 Specific inclusion and exclusion criteria for queries of the Danish registries**

| Phenotype | ICD-10 inclusion criteria | Age-specific inclusion criteria | Exclusion criteria |
|---|---|---|---|
| ASD | F84.0 | Diagnosis made at age 1 or later | None |
| SCZ | F20 | Diagnosis made at age 10 or later | None |
| ID | F70-F79 | Diagnosis made at age 1 or later | None |
| EP | G40.4C, G40.4E OR one of G40 with no additional specifications, G40.3 (any additional specification except G40.3F), G40.4 (any additional specification), G40.8, G40.9 with no additional specifications AND one of F70-F73, F78, F79, F84 with no additional specifications, F84.0, F84.8, F84.9 | Epilepsy diagnosis made by age 18 | Anyone with unknown parent, anyone with a parent who has any ICD-10 or ICD-8 epilepsy diagnosis, anyone with ICD-10 P code, or any of the following diagnostic codes recorded earlier than 1 year following the epilepsy diagnosis: A17, A39, A80-A89, C70, C71, E70-72, E74-80, E83, E85, E88, G00-G09 |
| CHD | All ICD codes listed in Table 3, except Q21.1 | Diagnosis made within 1 year of birth | one |

For each disorder, we describe the ICD-10 codes used to query the Danish data for comparable cases, as well as any age of diagnosis restrictions we applied to the data. We also describe the exclusion criteria applied to epilepsy cases

**Table 3 Aligning congenital heart disease phenotypes**

| Class of congenital heart disease | ICD-10 codes | N PGCG probands | N cases in Danish registry with fathers 20-29 | N cases in Danish registry with fathers >39 |
|---|---|---|---|---|
| Single ventricle | Q20.4, Q22.6, Q23.4 | 511 | 45 | 12 |
| TGA | Q20.3, Q20.5 | 294 | 75 | 32 |
| Tetrology of Fallot | Q21.3 | 363 | 95 | 36 |
| CTD not TGA | Q20.0, Q20.1, Q20.2, Q21.4 | 121 | 52 | 18 |
| Heterotaxy | Q20.6 | 38 | — | — |
| AVSD | Q21.2 | 50 | 107 | 32 |
| RVO/LVO | Q22.0, Q22.1, Q22.2, Q22.3, Q23.0, Q23.1, Q23.8, Q23.9, Q24.4, Q25.3 | 416 | 211 | 67 |
| Mv/TV | Q26.2, Q26.3 | 56 | 13 | 3 |
| Abnormal chamber | Q20.8, Q20.9, Q21.8, Q21.9, Q24.2 | 8 | 44 | 11 |
| VSD | Q21.0 | 119 | 873 | 220 |
| Vascular anomaly | Q24.5, Q25.1, Q25.2, Q25.4, Q25.5, Q25.6, Q25.7, Q25.8, Q25.9, Q26.0, Q26.1, Q26.2, Q26.3, Q26.8, Q26.9 | 187 | 200 | 52 |
| ASD | Q21.1 | 171 | — | — |
| Other | None | 14 | — | — |

"Class of congenital heart disease" refers to broad classes corresponding to Fyler codes found within the PCGC probands. These are ranked in order of clinical presentation from most to least severe. ICD-10 codes representing corresponding presentations were used to query the Danish registry data for each class of CHD. Trio probands refer to the number of probands whose CHD class is the most severe CHD in that person. Cases within the Danish registry refer to numbers of individuals who were given a corresponding ICD-10 code within one year of birth from each respective paternal age category. (See Supplementary Note 20 for an explanation as to why there are no cases listed under "heterotaxy", "ASD", or "other"). TGA transposition of the great arteries, CTD conotruncal defect, AVSD atrioventricular septal defect RVO/LVO ventricular outflow obstruction, MV/TV mitral or tricuspid valve anomaly, VSD ventricular septal defect, ASD atrial septal defect

For ASD, the only requirement for being a case was having the F84.0 ICD-10 diagnostic code assigned at age 1 or later. Our reason for this requirement is the assumption that any earlier diagnosis was likely either made in error or subject to a great deal of uncertainty. In Supplementary Note 19, we describe a set of alternative analyses in which we use a more expansive definition of ASD (including F84.5, F84.8 and F84.9) and explain why using only F84.0 is preferred.

For ID, we also required that an appropriate diagnostic code (F70-F79) be assigned at age 1 or later. It should be noted that the DDD cohort, which contributes a majority of the cases from which we derive our estimate for the dnSNV-only paternal age risk for ID, includes a small number of probands who themselves were not known to meet criteria for ID. However, there is no unifying phenotype within this cohort and a large majority are cognitively impaired (at least 87% in the portion of the cohort included in an earlier 2015 publication)[21]. Even those within the DDD cohort who had no recorded evidence of cognitive impairment were ascertained for having a significant neurodevelopmental disorder. Thus, we are confident that the overwhelming majority of trio probands used to estimate the dnSNV-mediated paternal age effect for ID were, in fact, cognitively impaired.

For SCZ, a case was defined as individual given a F20 diagnostic code at age 10 or later.

The probands ascertained into the trio-sequenced cohort investigating "neurodevelopmental disorders with epilepsy" all have syndromes that include

epilepsy as well as additional evidence of a serious neurodevelopmental disorder (almost always including intellectual disability)[24]. Our inclusion criteria for EPI within the Danish registry therefore required an additional diagnosis of a cognitive or neurodevelopmental disorder. The exception to this was specific epilepsy syndromes that always include additional signs of a serious neurodevelopmental problem (G40.4C and G40.4E). To be further consistent with the phenotypes included in the sequencing studies, the vast majority (if not all) of whom were diagnosed with epilepsy in childhood, we required that the epilepsy be diagnosed prior to age 18. Following the criteria for being included in the sequenced cohort, we also excluded any proband whose parent had an epilepsy diagnosis as well as any proband with an additional ICD code suggesting a likely infectious or traumatic etiology to their neurodevelopmental syndrome (Table 2).

The complete list of ICD codes used to diagnose CHD are listed in Table 3. This list was made using Filer codes associated with each case of CHD from the whole exome sequencing data and identifying the ICD codes that most closely align to the class of CHD captured by these codes. The one exception is that we did not query the Danish registry data for Q21.1 (corresponding to atrial septal defects). The reason for that is that atrial septal defects among the whole-exome sequenced CHD cases would, as a group, be substantially more severely affected than individuals in the Danish registry who were diagnosed with Q21.1. One such diagnosis had to be made prior to age 1 for an individual to be counted as a

case in this study. Because different forms of congenital heart disease may have different genetic architectures and therefore different degrees of risk associated with de novo variation, we performed an alternative analysis to investigate the epidemiologic association between paternal age and CHD under conditions where the distribution of CHD types is similar to that seen in the probands of the trio families included in this study. This alternative analysis is described in Supplementary Note 20.

## Data availability

The de novo variant rate data used in this study can be found in the main text or supplementary material of references[18–33]. The individual level data from the SSC cohort is available through application to the Simons Foundation. All other relevant data are contained within the article and its supplementary information or upon reasonable request.

## Code availability

All analyses for this study were performed using R. Custom code is made available at https://github.com/Jacob-L-Taylor/paternal-age.

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

## Acknowledgements

Our thanks to Rosy Hosking and Alice Galvin for their helpful reviews. Thanks to Alexander Frieden for his editorial comments. Thanks to the Pediatric Cardiac Genomics Consortia for their permission to use their data for analyses that took place prior to the publication of this data. E.B.R. was funded by National Institute of Mental Health grants R01MH111813 and K01MH099286-01A1, as well as Brain Behavior Research Foundation (NARSAD) Young Investigator grant 22379. J.L.T. and E.M.W. were funded by the Stanley Center for Psychiatric Research at the Broad Institute. J.P.G.D. was funded by the Danish Council for Independent Research DFF – 1331-00050. H.O.H. was supported by stipends from the Federal Ministry of Education and Research (BMBF), Germany, FKZ: 01EO1501 and the German Research Foundation (DFG): HE7987/1-1. C.E.S. is funded by the Howard Hughes Medical Institute. J.G.S. is funded by the Cardiovascular Development Consortium (1 U01 HL098166). Clinical information and exome data on CHD families was provided by the Pediatric Cardiac Genomics Consortium (U01-HL098188, U01-HL098147, U01-HL098153, U01-605 HL098163, U01-HL098123 and U01-HL098162). We also thank the families who took part in the Pediatric Cardiac Genomics Consortium and the Simons Simplex Collection study and the clinicians who collected data at each of the study sites. The iPSYCH project is funded by the Lundbeck Foundation and the universities and university hospitals of Aarhus and Copenhagen.

## Author contributions

Authors J.L.T., J.P.G.D., J.T.L., E.A., J.J.M., P.B.M., L.P., S.U.M., J.H., J.S., C.S., E.M.W., J.A.K., D.J.W., H.O.H., D.L., D.P.H., and E.B.R. contributed to the analyses. Authors J.L.T., A.B., M.J.D., and E.B.R. contributed to the design of the study. Authors J.L.T. and E.B.R. wrote the manuscript.

## Additional information

**Competing interests:** The authors declare no competing interests.

