## [Peer Review File · Nature Communications]

Reviewers' Comments:

Reviewer #1:

Remarks to the Author:

The manuscript from Taylor et al. is an elegant attempt to determine the proportion of paternal age-associated risk of five developmental and psychiatric disorders that can be attributed to age-related de novo mutations. This is an important question and (in my opinion) the results presented by Taylor and colleagues are interesting and novel.

The authors begin by developing a model of disorder risk associated with nonsynonymous dnSNVs, taking into account (i) the population relationship between paternal age and the rate of dnSNVs and (ii) the disorder-specific enrichment of dnSNVs in cases vs controls. They then use epidemiologic data from Denmark on paternal-age associated risk of these disorders to estimate the proportion of epidemiologic risk due to age-related mutations. They conclude that the epidemiologic risk is consistent with a mechanism of paternal age-related dnSNVs for CHD, ID and EPI but not ASD and SCZ.

The study design is elegant and displays admirable clarity of thought, but there are several limitations. The first is that the case cohorts for SCZ, ID, EPI and CHD do not have matched control data, and thus the SSC (ASD) sibling data is used as control data for all five disorders. The second is that parental origin of dnSNVs is unknown in the datasets used in the analysis, so there is potential for confounding due to the contribution of maternal mutations.

The authors are cognizant of these issues (and a few others) and go to some lengths to address them. To tackle the problem of technical differences between non-ASD case cohorts and the SSC sibling controls (which have the potential to bias estimates of the ORs due to dnSNVs) the authors adjusted the nonsynonymous dnSNV rates by a factor that makes the rate of synonymous dnSNVs equivalent. This is a nice solution, and perhaps better than the second (Samocha sequence context model) alternative approach, given the inability to calculate SE's. The authors argue that synonymous dnSNVs may contribute to trait variation by influencing gene expression and subsequently that "adjusting for rate of synonymous variation could subtly bias our conclusions about the marginal impact of nonsynonymous dnSNVs toward the null and could therefore cause us to slightly under-estimate the paternal age effect across disorders." I accept that this is a possibility, but there's currently little or no evidence for a role of synonymous dnSNVs in these disorders, and it would seem (to this referee) much more likely that failing to adjust for the synonymous rate, as a means of correcting for inevitable technical differences between cohorts, would result in biased estimates of the OR. For this reason, I would advocate presenting the adjusted results as the primary outcome. The authors suggest that results from the two approaches (raw vs synonymous-adjusted) are qualitatively the same, but technically the dnSNV vs epidemiologic estimates are no longer different for SCZ ($p=0.02$), after correcting for 5 tests.

A second way in which the use of SSC control data might influence estimates of the dnSNV-related OR is through differences in paternal age between each of the case samples and the SSC siblings. To evaluate this problem, the authors perform a within-family analysis for ASD in the SSC and show that paternal age differences between probands and sibs have minimal effect on estimates of the OR. This is convincing for ASD but given that paternal age is unknown for the other case cohorts (ID, EPI, CHD, SCZ), it's unclear to what extent this issue might influence the conclusions for other disorders? i.e. what would be the expected impact if the mean paternal age difference between ID cases and SSC siblings was significantly greater than that between SSC probands and siblings?. Given that adjusting for the synonymous rate might help to correct for any difference in mean paternal age between cohorts, this would be another strong justification for presenting the adjusted results as the primary output.

With respect to ASD, did the authors consider a model with both synonymous dnSNV adjustment and correction for paternal age differences (i.e. within families)?

The other key limitation is the inability to correct for maternal age. The author's acknowledge this in the Methods but (in my opinion) the issue is worthy of a paragraph in the Discussion. Some additional consideration beyond the points raised by the authors may also be warranted; for instance, what would be the likely impact on the inference made if the proportional rate of increase in dnSNVs differed between men and women? i.e. as seems possible given recent evidence for different mutational processes in men and women based on analysis of phased DNMs (Jonsson et al. 2017 Nature). What would be the likely impact if the rate of increase of dnSNVs was non-linear in women, as at least one study has suggested (Wong et al 2016)?

Minor comments:

Page 4: The sentence "Clarity regarding the amount of..." would make more sense if it read "Clarity regarding the proportion of paternal age-associated risk due to age-related de novo mutations...".

Table 1: Following from comments above, it would help to have the synonymous dnSNV rates included here, together with the adjusted NS dnSNV rates and raw and adjusted ORs.

Supp page 3 (end of first paragraph): "..., it is likely that for each..."

Supp page 7: (second paragraph, two instances): "...the mean number of dnSNVs..."

Supp page 13 (last line): $2431 * 0.27 / 0.25$ doesn't equal 1935.

Supp page 25: "We then take the 250th smallest and 250th largest..."

Reviewer #2:

Remarks to the Author:

The article considers the extent to which the observed associations between advanced paternal age and several disorders can be attributable to age-related increase in de novo mutations in male germline. The authors compare the odds ratios for the discussed disorders from the epidemiological research, with their estimates of the odds that could be attributable to the de novo single-nucleotide variants (dnSNVs), comparing children of fathers ages 20-29 and over 40. Significant differences between those estimates are interpreted as evidence for the paternal age effect being attributable to factors other than dnSNVs themselves.

The question itself is important to explore, and the paper provides a useful contribution to the field after Gratten et al. (2016) paper. Although the conclusions of those studies are largely similar (paternal age effect in neuropsychiatric disorders is not driven by the increase in germline dnSNVs), this work by Taylor et al. provides estimates computed using empirical data, as opposed to just simulations as in the earlier work.

Nevertheless, I feel there are several major issues that the authors need to address before the publication.

My first issue relates to the non-discussed differences in the etiological heterogeneity of the disorders selected by the authors (autism (ASD), schizophrenia (SCZ), congenital heart disease (CHD), intellectual disability (ID) and epilepsy (EPI)), which might have had an impact on the study conclusions. As shown in the earlier work (e.g. Iosiffov et al., 2014, Sanders et al., 2012), dnSNVs are relevant likely in only some (20-40%) proportion of cases of autism, with the remainder of cases likely explained mostly by the polygenic burden / environmental factors. This might not be equally applicable to ID, EPI and CHD. In other words, demonstrating that dnSNVs arising as the consequence of paternal ageing do not FULLY explain the epidemiological patterns

for ASD does not imply that for some proportion of cases they do, and the authors failed to acknowledge this point. The spike in the ASD prevalence in offspring of older men likely represents a mixture of contribution of both dnSNVs and higher ASD polygenic burden in men who delay fatherhood (review by Janecka et al., 2017, Mullins et al., 2017), and looking at either of those factors in isolation won't help in the context of a highly complex disorder. If the dnSNVs contribution tends to be more uniform in CHD, ID, EPI, the pattern of results would be exactly as the one reported by the authors. I think without addressing this issue, the authors cannot really draw the current conclusions.

My other issue is with the lack of consideration of non-random pattern of occurrence of the age-related dnSNVs. If they arise / propagate preferentially in autism-relevant genes (e.g. selfish spermatogonial selection model (Goriely et al., 2012)), then the first part of the algorithm, which is non-variable between the disorders, would be inaccurate. Gratten et al. showed that the selection model fits the simulated data well, so I was surprised the authors gave it no consideration here.

Next, the enrichment of dnSNVs (line 143) is concluded based on using the cohorts likely enriched for de novo mutations. I am not familiar with all of the cohorts cited by the authors, but at least Simons Simplex Collection (SSC) will have a larger proportion of cases where the disorder is attributable to de novo events than in the random sample of individuals with ASD. This should be discussed.

Related to this is another major issue, i.e. comparing de novo estimates in SSC with epidemiological observation from the Danish registry. Those cohorts would have completely different ascertainment procedures, resulting in very different sample composition - SSC is composed solely of the simplex cases, which are likely to be a minority among the cases sampled in the general population. In many places in the text the authors suggest that their dnSNVs results reflect something about paternal age effects in the population (e.g. lines 95, 188, 360, 376), and given the ascertainment bias, this does not seem warranted here.

Finally, the follow-up period in the Danish registry (1995-2012) seems very much delayed compared to the total ascertainment period (1955-2012). This could be an issue when investigating disorders with very different age of onset (~25 yrs in SCZ and <1yr in CHD), resulting in non-overlapping birth time intervals during which individuals with those conditions were sampled (e.g. a SCZ patient had to be born approximately 1970-1987; a CHD patient 1995-2012). Given changes in trends in the age at parenthood between those periods, the authors should at least acknowledge this as a limitation (or correct for the year of birth in the analyses).

Some other comments:

- 1) Line 66 - as men age, dnSNVs accumulate in their germline, not offspring
- 2) Line 138 - 3.1% of what? Do authors imply non-linear rate of accumulation of dnSNVs?
- 3) Line 190 - "inflated" does not seem to be an appropriate word here; the epidemiological estimates were indeed higher, but to the best of the authors' knowledge accurate
- 4) Line 218 - the claim is based on evidence pertaining to ASD only, yet the authors seem to suggest it's relevant to all complex disorders.
- 5) What is the authors' explanation for the rates of dnSNVs being similar in the current study to those reported previously in whole-genome sequencing work - in spite of known inflation of dnSNVs in exome methods (review by Segurel et al., 2014)? Could the controls in the SSC sample be a bit depleted for dnSNVs?
- 6) lines 384-5 - the method for calculating the impact of dnSNVs on the disorder seems to assume uniform penetrance / magnitude of the effect across the mutations (within-disorder). Is this a valid assumption to make?
- 7) line 420 - What do the authors mean by saying that their estimates are not valid for comparing estimates of dnSNVs within disorders?

Reviewer #3:

Remarks to the Author:

The authors developed a statistical model to evaluate whether the increase in 5 psychiatric and developmental disorders with paternal age can be explained by the nonsynonymous single nucleotide de novo mutations (dnSNV), which also increases with paternal age. They quantified the risk of these disorders due to increasing in de novo dnSNV with paternal age.

I think this is an important paper that for the first time we understand the contribution of dnSNVs on the psychiatric and developmental disorders. However, I feel that the authors try to convey the idea that de novo mutations due to advanced paternal age cannot explain the increase in incidence rate in epidemiology data. The last sentence in the discussion is a strong statement. It might well be true but one cannot draw such conclusion by looking at dnSNV only, there are many other types of variants and some may have a stronger effect.

1) We have known that CNVs play an important role in psychiatric and developmental disorders. Several recent papers have shown that the mutation rate of de novo SVs/CNVs increase with parental age, especially maternal age. But the authors failed to mention that CNVs could be explaining a larger proportion of the variance than dnSNV.

2. I also wonder if the authors can comment on the nonlinear relationship between paternal age and number of dnSNVs, does a nonlinear model explain the data better?

3. The study also allude to the possible correlation between having mutations that predispose oneself to the 5 disorders and the age at conception. And I believe they have enough data to quantify this risk, this would make the paper a lot more comprehensive.

Minor comments:

Figure 1 did not render well

A lot of values need to be properly formatted, e.g. $p=2e-10$ should be $P = 2 \times 10^{-10}$. This also applies to Table 1 etc.

I think this paper needs to be cited

Janecka M, Mill J, Basson MA, Goriely A, Spiers H, Reichenberg A, Schalkwyk L, Fernandes C. Advanced paternal age effects in neurodevelopmental disorders-review of potential underlying mechanisms. *Transl Psychiatry*. 2017 Jan 31;7(1):e1019.

Line 162, "children born to fathers 20-29 v >3": do you mean "versus" instead of "v"?

Supplementary Table 3: end of the paragraph:

987. Supplementary M, does that mean "supplementary material"?

Reviewers' comments:

Reviewer #1 (Remarks to the Author):

The manuscript from Taylor et al. is an elegant attempt to determine the proportion of paternal age-associated risk of five developmental and psychiatric disorders that can be attributed to age-related de novo mutations. This is an important question and (in my opinion) the results presented by Taylor and colleagues are interesting and novel.

The authors begin by developing a model of disorder risk associated with nonsynonymous dnSNVs, taking into account (i) the population relationship between paternal age and the rate of dnSNVs and (ii) the disorder-specific enrichment of dnSNVs in cases vs controls. They then use epidemiologic data from Denmark on paternal-age associated risk of these disorders to estimate the proportion of epidemiologic risk due to age-related mutations. They conclude that the epidemiologic risk is consistent with a mechanism of paternal age-related dnSNVs for CHD, ID and EPI but not ASD and SCZ.

The study design is elegant and displays admirable clarity of thought, but there are several limitations. The first is that the case cohorts for SCZ, ID, EPI and CHD do not have matched control data, and thus the SSC (ASD) sibling data is used as control data for all five disorders. The second is that parental origin of dnSNVs is unknown in the datasets used in the analysis, so there is potential for confounding due to the contribution of maternal mutations.

The authors are cognizant of these issues (and a few others) and go to some lengths to address them. To tackle the problem of technical differences between non-ASD case cohorts and the SSC sibling controls (which have the potential to bias estimates of the ORs due to dnSNVs) the authors adjusted the nonsynonymous dnSNV rates by a factor that makes the rate of synonymous dnSNVs equivalent. This is a nice solution, and perhaps better than the second (Samocha sequence context model) alternative approach, given the inability to calculate SE's. The authors argue that synonymous dnSNVs may contribute to trait variation by influencing gene expression and subsequently that "adjusting for rate of synonymous variation could subtly bias our conclusions about the marginal impact of nonsynonymous dnSNVs toward the null and could therefore cause us to slightly under-estimate the paternal age effect across disorders." I accept that this is a possibility, but there's currently little or no evidence for a role of synonymous dnSNVs in these disorders, and it would seem (to this referee) much more likely that failing to adjust for the synonymous rate, as a means of correcting for inevitable technical differences between cohorts, would result in biased estimates of the OR. For this reason, I would advocate presenting the adjusted results as the primary outcome. The authors suggest that results from the two approaches (raw vs synonymous-adjusted) are qualitatively the same, but technically the dnSNV vs epidemiologic estimates are no longer different for SCZ ($p=0.02$), after correcting for 5 tests.

We thank the reviewer for his or her careful attention to the manuscript, and thoughtful review. We have adopted the reviewer's suggestion to present, as our primary outcome, the dnSNV rates adjusted by synonymous *de novo* variant rate. These changes are reflected in an additional paragraph at the end of the introductory section, two additional sentences in the online methods, and a more extensive re-working of several sections of the Supplementary Note (a new section on "Adjusting for Synonymous Variation", modifications of the section now titled "Alternative analyses in which we do not adjust for rate of synonymous variation", and changes in the section on "Generating Confidence Intervals). This change is also reflected in

Table 1; Figures 2 and 3; Supplementary Tables 2, 3, 9 and 10; and Supplementary Figures 2 and 3.

The adjustment also led to some subtle changes in estimates for the dnSNV-driven paternal age effects and it widened the confidence intervals around our estimates. As a consequence, it introduced greater statistical ambiguity into whether our model suggests that the epidemiologic associations between advanced paternal age and ID/SCZ can be accounted for by dnSNVs (the p-value for the comparison in SCZ goes from 10^{-4} to 0.02; the p-value for the comparison in ID goes from 0.24 to 0.05). We address this with an additional paragraph in the results section:

“The association between advanced paternal age and ID risk in the Danish population may exceed that which can be accounted for by dnSNVs ($p = 0.05$). Minor variation in the statistical comparison is induced by control for synonymous rate, as described in the Supplementary Note (Supplementary Note and Supplementary Tables 9 and 10).”

Our major conclusions did not change:

- 1) The model still shows that dnSNVs that accumulate with increased paternal age are not likely to cause a major increase in disease risk. In the earlier version of this paper, we stated that it appears that across the genetically complex disorders investigated, an increase in paternal age from mid-20s to mid-40s would be expected to increase risk between 4-30%. In the current version, this changes to between 10-20%.
- 2) The evidence continues to suggest that dnSNVs alone are unlikely to explain epidemiologic associations between advanced paternal age and ASDs and SCZ. Whereas the available evidence suggests they could plausibly explain the relationship between advanced paternal age and CHD, EPI and ID.

A second way in which the use of SSC control data might influence estimates of the dnSNV-related OR is through differences in paternal age between each of the case samples and the SSC siblings. To evaluate this problem, the authors perform a within-family analysis for ASD in the SSC and show that paternal age differences between probands and sibs have minimal effect on estimates of the OR. This is convincing for ASD but given that paternal age is unknown for the other case cohorts (ID, EPI, CHD, SCZ), it's unclear to what extent this issue might influence the conclusions for other disorders? i.e. what would be the expected impact if the mean paternal age difference between ID cases and SSC siblings was significantly greater than that between SSC probands and siblings?. Given that adjusting for the synonymous rate might help to correct for any difference in mean paternal age between cohorts, this would be another strong justification for presenting the adjusted results as the primary output.

We appreciate the reviewer's argument and, as described above, have adjusted the primary analysis for synonymous rate. We added a sentence to Supplementary Note section “The model is robust to plausible variation in average effect size of dnSNVs”:

“However, because we adjusted for the rate of *de novo* synonymous variation across all disorders, any systematic differences in parental ages between the cohorts would be unlikely to distort the results.”

With respect to ASD, did the authors consider a model with both synonymous dnSNV adjustment and correction for paternal age differences (i.e. within families)?

We ran the model and added a sentence to the supplementary note accordingly: "We also ran a model in which we additionally included number of synonymous variants. This had no effect on the estimated impact of non-synonymous variants (beta of 0.1707 with SE of 0.0366 to 0.1708 with same SE)." Because the change made such little difference we do not show these results in Supplementary Figure 1.

The other key limitation is the inability to correct for maternal age. The author's acknowledge this in the Methods but (in my opinion) the issue is worthy of a paragraph in the Discussion. Some additional consideration beyond the points raised by the authors may also be warranted; for instance, what would be the likely impact on the inference made if the proportional rate of increase in dnSNVs differed between men and women? i.e. as seems possible given recent evidence for different mutational processes in men and women based on analysis of phased DNMs (Jonsson et al. 2017 Nature). What would be the likely impact if the rate of increase of dnSNVs was non-linear in women, as at least one study has suggested (Wong et al 2016)?

We moved the discussion of maternal age effects from online methods into the discussion (still with a call-out to section in the Supplementary Note). We added several sentences to acknowledge the issues suggested by the reviewer.

"the modeling approach used here results in estimates of paternal-age risk that include (and are increased by) the correlated risks associated with maternal age. As the model is designed to predict changes in disease incidence at a population-level, the results presented here will best generalize to populations in which maternal and paternal age have a similar relationship to that observed in the United States."

Minor comments:

Page 4: The sentence "Clarity regarding the amount of..." would make more sense if it read "Clarity regarding the proportion of paternal age-associated risk due to age-related de novo mutations...".

-We made this suggested edit

Table 1: Following from comments above, it would help to have the synonymous dnSNV rates included here, together with the adjusted NS dnSNV rates and raw and adjusted ORs.

-Changed

Supp page 3 (end of first paragraph): "..., it is likely that for each..."

-Corrected (added the word "is")

Supp page 7: (second paragraph, two instances): "...the mean number of dnSNVs..."

-We made suggested changes (added the word "mean" twice)

Supp page 13 (last line): $2431 * 0.27 / 0.25$ doesn't equal 1935.

-This should have read 2625 (which reflects the analysis we actually did). This was fixed in the text.

Supp page 25: "We then take the 250th smallest and 250th largest..."

-Fixed (added the word "and")

Reviewer #2 (Remarks to the Author):

The article considers the extent to which the observed associations between advanced paternal age and several disorders can be attributable to age-related increase in de novo mutations in male germline. The authors compare the odds ratios for the discussed disorders from the epidemiological research, with their estimates of the odds that could be attributable to the de novo single-nucleotide variants (dnSNVs), comparing children of fathers ages 20-29 and over 40. Significant differences between those estimates are interpreted as evidence for the paternal age effect being attributable to factors other than dnSNVs themselves.

The question itself is important to explore, and the paper provides a useful contribution to the field after Gratten et al. (2016) paper. Although the conclusions of those studies are largely similar (paternal age effect in neuropsychiatric disorders is not driven by the increase in germline dnSNVs), this work by Taylor et al. provides estimates computed using empirical data, as opposed to just simulations as in the earlier work.

Nevertheless, I feel there are several major issues that the authors need to address before the publication.

My first issue relates to the non-discussed differences in the etiological heterogeneity of the disorders selected by the authors (autism (ASD), schizophrenia (SCZ), congenital heart disease (CHD), intellectual disability (ID) and epilepsy (EPI)), which might have had an impact on the study conclusions. As shown in the earlier work (e.g. Iosifov et al., 2014, Sanders et al., 2012), dnSNVs are relevant likely in only some (20-40%) proportion of cases of autism, with the remainder of cases likely explained mostly by the polygenic burden / environmental factors. This might not be equally applicable to ID, EPI and CHD. In other words, demonstrating that dnSNVs arising as the consequence of paternal ageing do not FULLY explain the epidemiological patterns for ASD does not imply that for some proportion of cases they do, and the authors failed to acknowledge this point. The spike in the ASD prevalence in offspring of older men likely represents a mixture of contribution of both dnSNVs and higher ASD polygenic burden in men who delay fatherhood (review by Janecka et al., 2017, Mullins et al., 2017), and looking at either of those factors in isolation won't help in the context of a highly complex disorder. If the dnSNVs contribution tends to be more uniform in CHD, ID, EPI, the pattern of results would be exactly as the one reported by the authors. I think without addressing this issue, the authors cannot really draw the current conclusions.

We appreciate the reviewer's attention to heterogeneity. Across all the disorders studied, there is enormous heterogeneity in etiology. Genetic and environmental contributions are relevant to each; and the genetic factors relevant to each disorder are highly heterogeneous. Our approach focuses on the question of: "what fraction of cases have a risk-contributing de novo mutation?" For each of the 5 disorders, the answer to that question is some but not all. We use results from recent sequencing studies to note the percentage of cases of each disorder that carry a de novo mutation in the exome. For intellectual disability, the percentage is high (but is still far less than 100). For schizophrenia and autism it is much lower. We expect and specifically model the fact that many to most cases of each of these disorders will not have a contributing de novo mutation.

My other issue is with the lack of consideration of non-random pattern of occurrence of the age-related dnSNVs. If they arise / propagate preferentially in autism-relevant genes (e.g. selfish

spermatogonial selection model (Goriely et al., 2012)), then the first part of the algorithm, which is non-variable between the disorders, would be inaccurate. Gratten et al. showed that the selection model fits the simulated data well, so I was surprised the authors gave it no consideration here.

We have added a section in the Supplementary Note (which includes a novel empirical analysis of the SSC families) to clarify the relationship between the selfish spermatogonial selection (SSS) model and our study. In the main text, we have included Goriely et al. 2012 as a reference, and note that we do not believe that SSS would substantially influence our findings.

As now discussed in the supplementary note, we respectfully disagree with the reviewer's characterization of Gratten et al.'s simulation study as "showing that the selection model fits the simulated data well." In their results section Gratten et al (Nat Genetics 2016) state:

"In model 2, which considers paternal-age-related mutations together with selfish selection. . .the additional risk due to selfish selection in older fathers is trivial for most plausible combinations of the parameters."

In their discussion, they state:

"Currently, the only empirical evidence for selfish selection comes from studies of rare monogenic disorders such as Apert syndrome. Our results suggest that it is unlikely that this mechanism makes a major contribution to the increased risk of common complex psychiatric disorders in children of older men."

Next, the enrichment of dnSNVs (line 143) is concluded based on using the cohorts likely enriched for de novo mutations. I am not familiar with all of the cohorts cited by the authors, but at least Simons Simplex Collection (SSC) will have a larger proportion of cases where the disorder is attributable to de novo events than in the random sample of individuals with ASD. This should be discussed.

The relationship between de novo rate in the SSC and that of a random sample of people with ASDs (which would be very difficult to ascertain!) is unclear. The de novo rate in the SSC is fairly average compared to many other ASD cohorts, despite it being a simplex collection. The SSC's exclusion criteria were far stronger than many other ASD cohorts (e.g. AGRE), and the leaderships' decision to exclude ASD individuals with "syndromic features" likely substantially reduced the de novo rate in the sample.

We take the reviewers point that clinically ascertained cohorts are non-random, and have noted a point to that effect in the discussion.

"Similarly, none of the cohorts used are random samples of individuals diagnosed with specific disorders. It is possible that the different ascertainment strategies used for each cohort lead to subtly different rates of dnSNVs. "

Related to this is another major issue, i.e. comparing de novo estimates in SSC with epidemiological observation from the Danish registry. Those cohorts would have completely different ascertainment procedures, resulting in very different sample composition - SSC is composed solely of the simplex cases, which are likely to be a minority among the cases

sampled in the general population. In many places in the text the authors suggest that their dnSNVs results reflect something about paternal age effects in the population (e.g. lines 95, 188, 360, 376), and given the ascertainment bias, this does not seem warranted here.

We agree that the ascertainment methods in the trio studies likely leads to cases that differ in certain relevant respects from cases ascertained through the Danish registries. Specifically, we agree that cases from the trio studies are, if anything, likely to have a higher *de novo* burden compared with cases in any general population. Therefore, we would expect our results from the *de novo* only model to possibly *overestimate* the role of *de novo* variants in driving the observed epidemiologic associations. However, we believe that the possibility of this bias actually strengthens our argument that much of the observed association between advanced paternal age and neuropsychiatric illness, is likely due to other causes. We added a sentence to the discussion addressing this point:

“Thus, it seems plausible that the true differences between the paternal age effect observed in the Danish population and the impact due to paternal-age related dnSNVs may be larger than suggested by Figure 3.”

Finally, the follow-up period in the Danish registry (1995-2012) seems very much delayed compared to the total ascertainment period (1955-2012). This could be an issue when investigating disorders with very different age of onset (~25 yrs in SCZ and <1yr in CHD), resulting in non-overlapping birth time intervals during which individuals with those conditions were sampled (e.g. a SCZ patient had to be born approximately 1970-1987; a CHD patient 1995-2012). Given changes in trends in the age at parenthood between those periods, the authors should at least acknowledge this as a limitation (or correct for the year of birth in the analyses).

We address this problem for CHD by limiting controls to those born between 1994-2011 (the same time period during which cases of CHD had to be born).

For the other disorders, we did a set of analyses adjusting for calendar time of individual birth (we mentioned this in online methods, described the analyses in the Supplementary Note and report the results in Supplementary Table 6.) For SCZ, EP, and ID, these analyses did not substantively change the results, as presented in Figure 3. For ASD, the estimate of the OR associated with advanced paternal age decreased from 1.68 to 1.45 when adjusting for calendar time. However, even with this decrease, the difference with the dnSNV model remains highly statistically significant.

Some other comments:

1) Line 66 - as men age, dnSNVs accumulate in their germline, not offspring

We have rewritten this sentence to read: “As men age, dnSNVs are observed to increase in their offspring.”

2) Line 138 - 3.1% of what? Do authors imply non-linear rate of accumulation of dnSNVs?

We have added an additional section to the supplementary note to explain our choice to model the relationship between paternal age and number of dnSNVs using a Poisson model, rather than a linear model. In short, modeling this relationship proportionally allows us, mathematically, to treat all dnSNVs en masse, rather than attempting to break out different classes of dnSNVs

that impact each disorder differentially. We show in new Supplementary Figures (5a and 5b) that either a Poisson model or a linear model, appears to match the data well.

3) Line 190 - "inflated" does not seem to be an appropriate word here; the epidemiological estimates were indeed higher, but to the best of the authors' knowledge accurate

We have changed "inflated" to "larger" to remove the unintended implication that the epidemiological estimates are not accurate

4) Line 218 - the claim is based on evidence pertaining to ASD only, yet the authors seem to suggest it's relevant to all complex disorders.

We added the words "of which we are aware" so the sentence now reads "there is currently no evidence of which we are aware for enrichment of de novo variants outside of the exome in genetically complex human diseases."

5) What is the authors' explanation for the rates of dnSNVs being similar in the current study to those reported previously in whole-genome sequencing work - in spite of known inflation of dnSNVs in exome methods (review by Segurel et al., 2014)? Could the controls in the SSC sample be a bit depleted for dnSNVs?

The SSC control sample is not large enough to say whether the dnSNV rate is significantly lower than expectation.

6) lines 384-5 - the method for calculating the impact of dnSNVs on the disorder seems to assume uniform penetrance / magnitude of the effect across the mutations (within-disorder). Is this a valid assumption to make?

The model is specifically built to accommodate effect size heterogeneity of dnSNVs within each disorder. We have clarified that point within the methods and supplementary note.

7) line 420 - What do the authors mean by saying that their estimates are not valid for comparing estimates of dnSNVs within disorders?

We added a detailed explanation to the Supplemental Note:

"The confidence intervals illustrate the range of possibilities for each comparison (e.g. offspring ASD risk between fathers aged 45 v. 25). However, some of the same parameters are used to generate estimates within and across disorders. Therefore, these estimates are not independent of one another and the associated confidence intervals are not valid for comparing estimates of the dnSNV to one another. For example, Figure 1 makes it appear that the estimate for offspring ASD risk between fathers aged 45 v. 25 is substantially overlapping the risk for fathers aged 55 v. 25. However, our confidence that the risk for offspring of 55 year-olds is higher than the risk for offspring of 45 year-olds is actually quite high, as it is a function of our confidence that more dnSNVs lead to more risk for ASD ($p = 3e-8$) and our confidence that dnSNVs increase with older paternal age ($p = 2e-10$)."

Reviewer #3 (Remarks to the Author):

The authors developed a statistical model to evaluate whether the increase in 5 psychiatric and developmental disorders with paternal age can be explained by the nonsynonymous single nucleotide de novo mutations (dnSNV), which also increases with paternal age. They quantified

the risk of these disorders due to increasing in de novo dnSNV with paternal age.

I think this is an important paper that for the first time we understand the contribution of dnSNVs on the psychiatric and developmental disorders. However, I feel that the authors try to convey the idea that de novo mutations due to advanced paternal age cannot explain the increase in incidence rate in epidemiology data. The last sentence in the discussion is a strong statement. It might well be true but one cannot draw such conclusion by looking at dnSNV only, there are many other types of variants and some may have a stronger effect.

We appreciate the reviewer's comment and have edited the last paragraph to avoid arguments beyond the evidence presented in the manuscript.

We have added the words "accumulating dnSNVs through" so the last sentence now reads: "In fact, our data suggest that an incidence increase of this magnitude would only result from accumulating dnSNVs through delayed childbearing if American men were, on average, now conceiving their children when well over 100 years old (Supplementary Note)."

1) We have known that CNVs play an important role in psychiatric and developmental disorders. Several recent papers have shown that the mutation rate of de novo SVs/CNVs increase with parental age, especially maternal age. But the authors failed to mention that CNVs could be explaining a larger proportion of the variance than dnSNV.

CNVs are important to consider and we briefly introduce the topic in the main text:

"As described in the Supplementary Note and Supplementary Figure 1, the model's estimates are robust to: plausible variation in the expected (control) rate of dnSNVs, plausible variation in the estimated effect size of case dnSNVs, **and the inclusion of de novo copy number variants.**" [bolded here for emphasis]

We provide a detailed discussion of this issue in the Supplementary Note and Supplementary Figure 1. We show that although dnCNVs do, on average, have a greater impact than dnSNVs and may also increase with advancing paternal age, the fact that dnCNVs are so much rarer strongly limits the possibility for their mediating much of the association between advanced paternal age and disorder risk.

2. I also wonder if the authors can comment on the nonlinear relationship between paternal age and number of dnSNVs, does a nonlinear model explain the data better?

Please see response to reviewer #2 above.

3. The study also allude to the possible correlation between having mutations that predispose oneself to the 5 disorders and the age at conception. And I believe they have enough data to quantify this risk, this would make the paper a lot more comprehensive.

We introduce the hypothesis that shared common genetic variation (i.e. the type of variation identified through genome-wide association studies) may underlie both a) risk for certain neuropsychiatric disorders and b) delayed parenthood. In doing so, we cite empirical work that supports the hypothesis (Mehta et al, 2016; Barban et al 2016).

The data used in this study do not allow us to quantify or estimate this phenomenon. It would require samples that are orders of magnitude larger than we have on hand for this project, GWAS data (which we do not use in this project). It is a very important question, but one that we feel is outside the scope of this project.

Minor comments:

Figure 1 did not render well

Thank you – we have tried a new upload approach for the resubmission and will adjust further as necessary.

A lot of values need to be properly formatted, e.g. $p=2e-10$ should be $P = 2 \times 10^{-10}$. This also applies to Table 1 etc.

We have made the change.

I think this paper needs to be cited

Janecka M, Mill J, Basson MA, Goriely A, Spiers H, Reichenberg A, Schalkwyk L, Fernandes C. Advanced paternal age effects in neurodevelopmental disorders-review of potential underlying mechanisms. *Transl Psychiatry*. 2017 Jan 31;7(1):e1019.

We have cited this paper after first sentence.

Line 162, "children born to fathers 20-29 v >3": do you mean "versus" instead of "v"?

We wrote out "versus" in text.

Supplementary Table 3: end of the paragraph:

987. Supplementary M, does that mean "supplementary material"?

We have changed to "the Supplementary Note."

Reviewers' Comments:

Reviewer #1:

Remarks to the Author:

The authors have addressed all of the concerns I raised in my review.

Reviewer #2:

Remarks to the Author:

The Authors have put considerable effort into improving the manuscript, and I was glad to see that they addressed, to some extent, all the comments I raised previously. The study represents a valuable contribution to the existing literature, nevertheless, there remain some issues that still need further clarification.

Upon reading the article again, it seems that the timescale used for those calculations did not account for the fact that the accrual of the mutations is only expected to begin after puberty (and thus fathers aged 50 had more than double the time to accrue those mutations, compared to fathers aged 25 - unlike what is assumed in the model currently used in the manuscript). Poisson regression should easily accommodate different rates for different time intervals, so given the biological plausibility, there seems to be no reason for not correcting for this? If I'm mistaken, the issue should be at least explicitly clarified in the manuscript.

It was good to see that the authors adjusted for the calendar year, and included those estimates next to those from models adjusted for maternal age. However, there seems to be little rationale for not controlling for both in the same model?

Supplementary Table 6 suggests the patterns are quite different for ASD than the other disorders (the only dx sensitive to calendar year, and for which ORs decrease when controlling for maternal age), and I don't think this should be buried on page 55 of the Supplementary Material. Knowing how sensitive the epidemiological estimates are to both calendar year and maternal age, showing unadjusted ASD odds in Figure 3, and thus a gaping hole between dnSNVs model and epidemiological patterns (for ASD, much less gaping after correcting for the calendar year alone), may be a bit misleading, especially if those effects are not proportional across the different disorders - especially when authors make a point of comparing those 5 disorders.

{and the "and/or", line 447 of the manuscript, does not appear to be accurate, given what it presented in S.table 6}

Finally, in my opinion more clarity is still needed with regard to the notion of etiological heterogeneity. It is well established that dnSNVs have differential contribution to the disorders investigated, and I think it may still be a bit misleading to present the findings without acknowledging this more explicitly (e.g. if, at baseline, dnSNVs contribute to a much smaller fraction of ASD than ID cases, one cannot be "surprised" paternal age-related increase in dnSNVs is more predictive of the PA-related increase of the prevalence of the latter outcome?). One suggestion would be to include a supplementary table/figure comparing the dnSNVs model to epidemiological estimates of childhood autism (severe end of ASD) in DK?

Minor:

Line 51 + 11 others - ASD should be singular

Line 204 - I doubt clinicians can make use of these results

Line 242 - is there sufficient evidence to claim, at this point, that "dnSNVs (...) is the form of de novo mutation that is, but far, most strongly associated with paternal age"? My feeling is that

predominantly, this is simply the one that has been studied most.

Lines 241- 247 - This is a scientifically solid study of 5 disorders, yet the final conclusion singles out ASD as the only focus. I see no reason to do that, not even when citing the New York Times. The final conclusion should follow from the specific goal of the study.

Reviewer #3:

Remarks to the Author:

The authors have sufficiently addressed my concerns and I have no further comments.

Thank you for the opportunity to revise and resubmit our manuscript for a second time. We were happy to see that Reviewer 1 and 3s queries were satisfied through the first revision. Please find below our response to Reviewer 2's second set of comments.

1. Upon reading the article again, it seems that the timescale used for those calculations did not account for the fact that the accrual of the mutations is only expected to begin after puberty (and thus fathers aged 50 had more than double the time to accrue those mutations, compared to fathers aged 25 - unlike what is assumed in the model currently used in the manuscript). Poisson regression should easily accommodate different rates for different time intervals, so given the biological plausibility, there seems to be no reason for not correcting for this? If I'm mistaken, the issue should be at least explicitly clarified in the manuscript.

We have addressed this question in a supplementary analysis. We changed the model such that the accumulation in de novo mutation rate begins at age 13 (an arbitrarily selected age proxy for puberty). We see no statistically significant or substantive difference in the estimated effect of paternal age when we make this change. Reducing the accumulation timescale reduces the standard errors on the accumulation effect estimate, which is reflected in minorly altered p-values in the modified analysis. Because a) accumulation may begin before puberty, b) the selection of age 13 (or any other age) is an arbitrary estimate of puberty onset, and c) we see no difference in effect or interpretation of the data, we have added this analysis to the supplement rather than the main text. We have added a sentence to the main text indicating that the issue was thoughtfully considered (see line 133: "Finally, in the Supplementary Note we show that the results are not substantially affected. . ."), and results to that effect can be found in the supplement (pages 18-20). We also added a column to Supplementary Table 3 showing how making this assumption would affect p-values with respect to the difference between our model's estimates and the Danish epidemiologic results.

2. It was good to see that the authors adjusted for the calendar year, and included those estimates next to those from models adjusted for maternal age. However, there seems to be little rationale for not controlling for both in the same model? Supplementary Table 6 suggests the patterns are quite different for ASD than the other disorders (the only dx sensitive to calendar year, and for which ORs decrease when controlling for maternal age), and I don't think this should be buried on page 55 of the Supplementary Material. Knowing how sensitive the epidemiological estimates are to both calendar year and maternal age, showing unadjusted ASD odds in Figure 3, and thus a gaping hole between dnSNVs model and epidemiological patterns (for ASD, much less gaping after correcting for the calendar year alone), may be a bit misleading, especially if those effects are not proportional across the different disorders - especially when authors make a point of comparing those 5 disorders. {and the "and/or", line 447 of the manuscript, does not appear to be accurate, given what it presented in S.table 6}

We've pasted the relevant contents of Supplementary Table 6, below.

Disorder	Crude HR/OR (95% CI)	HR/OR adjusted for maternal age (95% CI)	HR/OR adjusted for calendar time (95% CI)
SCZ	1.31 (1.23-1.40)	1.40 (1.29-1.51)	1.31 (1.23-1.40)
ASD	1.68 (1.51-1.86)	1.52 (1.34-1.71)	1.45 (1.30-1.61)
EP	1.39 (1.08-1.79)	1.52 (1.14-2.03)	1.36 (1.06-1.74)
ID	1.44 (1.33-1.56)	1.46 (1.33-1.61)	1.40 (1.29-1.52)
CHD	1.03 (0.92-1.14)	1.00 (0.89-1.13)	-

We note that there is no evidence for a statistically significant difference between the unadjusted and adjusted ORs in this table, including for ASD. While the point estimates may appear to vary, the confidence intervals are substantially overlapping. While this of course could reflect statistical power, it is not an issue we can resolve using the Danish population registry (and over 3 million person years). As we are not powered to detect a significant change in the OR for paternal age when controlling for either of the variable of interest, we feel that a model that includes both is not needed, and could create confusion as well as inflate our type 1 error rate. We have noted that there could be joint effects of calendar year and maternal age that we are not powered to see in the manuscript (online methods, line 483) and the supplement (bottom of page 35).

3. Finally, in my opinion more clarity is still needed with regard to the notion of etiological heterogeneity. It is well established that dnSNVs have differential contribution to the disorders investigated, and I think it may still be a bit misleading to present the findings without acknowledging this more explicitly (e.g. if, at baseline, dnSNVs contribute to a much smaller fraction of ASD than ID cases, one cannot be "surprised" paternal age-related increase in dnSNVs is more predictive of the PA-related increase of the prevalence of the latter outcome?). One suggestion would be to include a supplementary table/figure comparing the dnSNVs model to epidemiological estimates of childhood autism (severe end of ASD) in DK?

We agree with the reviewer that subsets of the ASD population will differ in terms of their PA-related increase in de novo risk (see, for example, Weiner et al., Nature Genetics 2017). From related research (e.g. Kosmicki et al, Nature Genetics 2017), we agree with the reviewer that the greater role of PA-related dnSNVs in ID compared to ASD is not surprising. However, it is perhaps surprising that the population estimates of paternal age risk do not track with the de novo estimates. We think it is interesting to point out that despite the greater etiologic importance of dnSNVs for ID compared with ASD, that the epidemiologic association between advanced paternal age seems to be at least as substantial for ASD as it is for ID.

To address the reviewer's concern that we need to do better at explicitly acknowledging the fact that dnSNVs contribute different amounts of risk for different disorders we add a sentence to the discussion (line 232): "It is possible that the different ascertainment strategies used for each cohort lead to subtly different rates of dnSNVs. This is particularly possible in the case of disorders with substantial phenotypic and etiologic heterogeneity, like ASD." We cannot pursue the analysis suggested by the reviewer for several reasons, including a) lack of data to that effect from the large ASD consortia used to generate the de novo estimates and b) limited diagnostic reliability between ASD subcategories, which is clear in analysis of registry data (many people have multiple ASD subtypes noted in the record; see Lord and Jones JCPP 2012).

Minor:

Line 51 + 11 others - ASD should be singular

We have made this change.

Line 204 - I doubt clinicians can make use of these results

We have changed the text to “genetic counselors and others” in lieu of “clinicians.”

Line 242 - is there sufficient evidence to claim, at this point, that "dnSNVs (...) is the form of de novo mutation that is, but far, most strongly associated with paternal age"? My feeling is that predominantly, this is simply the one that has been studied most.

We have removed that claim

Lines 241- 247 - This is a scientifically solid study of 5 disorders, yet the final conclusion singles out ASD as the only focus. I see no reason to do that, not even when citing the New York Times. The final conclusion should follow from the specific goal of the study.

We have removed that text.

Sincerely,

Elise Robinson and Jacob Taylor

Reviewers' Comments:

Reviewer #1:

Remarks to the Author:

The authors have satisfactorily addressed the additional concerns raised by Reviewer #2.